# Effect of Annealing on Surface Morphology and Structure of Nickel Coatings Deposited from Deep Eutectic Solvents

**Juliusz Winiarski** [1,*], **Anna Niciejewska** [1], **Włodzimierz Tylus** [1], **Katarzyna Winiarska** [2], **Karolina Pazgan** [1] **and Bogdan Szczygieł** [1]

[1] Group of Surface Technology, Department of Advanced Material Technologies, Faculty of Chemistry, Wrocław University of Science and Technology, Wybrzeże Wyspiańskiego 27, 50-370 Wrocław, Poland; anna.niciejewska@pwr.edu.pl (A.N.); wlodzimierz.tylus@pwr.edu.pl (W.T.); karolina.pazgan@wp.pl (K.P.); bogdan.szczygiel@pwr.edu.pl (B.S.)

[2] Division of Analytical Chemistry and Chemical Metallurgy, Faculty of Chemistry, Wrocław University of Science and Technology, Wybrzeże Wyspiańskiego 27, 50-370 Wrocław, Poland; katarzyna.winiarska@pwr.edu.pl

[*] Correspondence: juliusz.winiarski@pwr.edu.pl; Tel.: +48-713203193

**Abstract:** Nanocrystalline nickel coatings deposited on a copper base material from DES made of choline chloride and ethylene glycol in 1:2 molar ratio containing 1 mol $dm^{-3}$ $NiCl_2 \cdot 6H_2O$ were modified through post-deposition heat treatment at the temperatures from 100 to 400 °C. As-deposited coatings were composed of spheroidal agglomerates with the size of several hundred nanometers interspersed with lamellar crystals, but after annealing at 300 °C and 400 °C only single nano-sized plates embedded in a granular and porous layer remained. As the temperature of the heat treatment increased from 100 °C to 400 °C, the mean crystallite size increased from 13 to 35 nm. The change in crystallite size was accompanied by a change in microhardness, the maximum value of which was measured for the annealed coating at 200 °C. As a result of heat treatment, coatings were gradually covered by a layer of oxidized nickel species. XPS analyses showed that NiOOH and $Ni(OH)_2$ dominated among them. Above 200 °C the share of these compounds began to decline in the face of the increasing share of NiO. This, in turn, clearly translated into a deterioration of the corrosion resistance of Ni coatings annealed at 300 °C, and especially at 400 °C, during exposure in 0.05 mol $dm^{-3}$ NaCl solution.

**Keywords:** electron microscopy; microhardness; X-ray analysis; nanocrystalline materials; nickel coatings; grains and interfaces; porosities; heat treatment; corrosion resistance

## 1. Introduction

Non-aqueous plating baths based on deep eutectic solvents (DES) produce nanocrystalline nickel coatings with very low surface roughness and good corrosion resistance [1]. Moreover, as demonstrated by Abbott et al. [2] modification of the structure, phase composition, internal stresses and crystallite size of nickel coatings deposition from DES compared to those obtained from conventional Watt's baths can significantly affect their corrosion resistance, hardness and other mechanical properties. Another possible way to improve the performance of the already deposited nickel coatings is to apply additional heat treatment. De Los Santos Valladares et al. [3] proved that oxidation of nickel surface extended from the outermost layers towards bulk of the coating. As a product of this oxidation, a granular and nanospheroidal layer made of nickel and nickel oxide was formed. Finally, the resulting surface was amorphous with crystalline areas. Laszczyńska et al. [4] showed that exposure of Ni–Mo coatings to high temperature caused the formation of a nickel oxide layer. Full oxidation took place during heat treatment at 700 °C. At 350 °C, the degree of crystallization has increased despite the increase in NiO content. These changes affected thickness of this layer and its barrier properties when exposed to corrosive

environment [4]. Annealed nickel coatings (electrodeposited from baths based on DES) can be used as nickel oxide electrodes for the production of supercapacitors [5].

Thuvander et al. [6] have indicated that heat treatment at temperatures of 190–320 °C resulted in the abnormal grain growth. The huge number of grain boundaries in nanocrystalline material caused that the driving force of grain growth was significant even during annealing at low temperatures (200 °C). This caused changes in the properties of the material and thus its applicability. More precisely De Los Santos Valladares et al. [3] noticed that during annealing the roughness and porosity of nickel coatings increased, the density of the coatings decreased and the cracks reaching the substrate appeared. Lü et al. [7] indicates that the abnormal grain growth can be favored by sulfur and carbon contamination of the coating (originating from organic additives in plating bath, such as e.g., saccharin). During annealing, carbon and sulfur impurities migrate along the grain boundaries, preventing normal grain growth. On the other hand, the coatings obtained from the bath without organic additives do not show abnormal grain growth after heat treatment. Annealing at a temperature above 300 °C reduces the microhardness and residual stresses in the coating with a simultaneous significant increase in grain size [7]. Heat treatment can also affect the change in the preferred orientation of the crystallites in nickel coatings [7] and Ni–45% Fe alloy coatings [8].

Interestingly, annealing of nickel coatings does not only cause changes to the surface and the bulk of coating, but also, they may occur on the base material/coating interface. The formation of intermetallic layers between the base metal and the coating is primarily influenced by the duration of high temperature exposure [9]. These intermetallic layers can improve the mechanical properties and adhesion of the coatings [10]. Although the heat treatment influences coatings microhardness, the addition of an alloying component, e.g., tungsten, to the coating has an even greater impact on the above-mentioned property. This is a result of the phases formed, e.g., $Ni_4W$, $NiW$ [11]. Although Laszczyńska et al. [4] showed that the heat treatment caused an increase in microhardness of Ni−Mo coating, it was accompanied by an increase in its brittleness and deterioration of its adhesion to the steel substrate.

In the light of these reports, there are only few publications available on the annealing of Ni coatings deposited in DES baths. Therefore, our work was aimed at deepening the knowledge in this field. We investigate the effect of heat treatment temperature on the properties of nickel coatings electrodeposited from DES plating bath composed of (2-hydroxyethyl)trimethylammonium chloride and ethylene glycol (1:2 molar ratio) with the addition of $NiCl_2$ as the source of nickel ions. We believe that characterizing changes in the chemical composition of metal oxidation products is key to determining the behavior of the coating during exposure in an aggressive environment. To achieve this scientific goal surface morphology, as a function of heating temperature, was tracked by scanning electron microscopy (SEM). The chemistry of the surface of Ni coatings subjected to heat treatment was analyzed by X-ray photoelectron spectroscopy (XPS). Linear polarization resistance (LPR) technique and electrochemical impedance spectroscopy (EIS) were used to characterize the corrosion resistance of Ni coatings in 0.05 mol $dm^{-3}$ solution of NaCl. Coating's microstructure was analyzed by FIB/SEM technique, supported by X-ray diffraction (XRD), and finally correlated with the results of microhardness measurements.

## 2. Materials and Methods

### 2.1. Materials, Coatings Electrodeposition and Heat Treatment

DES solvent was synthesized by mixing choline chloride (≥98%, Sigma®, St. Louis, MO, USA) with ethylene glycol (Merck, EMD Milipore, Darmstadt, Germany), both "as supplied", in 1:2 molar ratio at 60 °C. After obtaining a transparent liquid, $NiCl_2 \cdot 6H_2O$ (POCH, p.p.a.) was added in a concentration of 1 mol $dm^{-3}$ and the bath was stirred at 60 °C until complete dissolution of nickel chloride. This plating bath was developed in preliminary tests [12] while the plating parameters were optimized more recently [13]. The

water content in the bath (7.4 wt.%) was determined by Karl–Fischer titration according to IEC 60814:1997.

Copper disks (M1E grade, geometric area 4.3 cm$^2$) were used as the base material. Copper was first mechanically polished (#600–#2000 abrasive paper), washed in DI water and degreased in methanol using the ultrasonic cleaner. During electrolysis the cathode was mounted between two nickel anodes (99.99% Ni) in a 100 mL vessel with thermostat jacket (Metrohm). Plating bath was stirred (IKA Toppolino, round PTFE-coated magnetic stirring bar 15 mm × 6 mm, 200 rpm). Electrodeposition parameters: a constant current density $j_c$ = 6 mA cm$^{-2}$, time 90 min, temperature 70 °C were selected basing on our exploratory research and previous publication [13]. Finally, samples were rinsed in DI water, methanol, dried in a vacuum desiccator and stored under Ar atmosphere. Ni coatings with an average thickness of 7–8 μm were obtained under the given parameters. The current efficiency, calculated on the basis of charge passed, mass gain and coatings cross-section measurements, was estimated at ca. 65–75%.

After electrodeposition the coatings were heat-treated in the oven at: 100, 200, 300 and 400 °C under air atmosphere (Czylok FCF-V20/mini furnace). The samples were heated with a step 5 °C min$^{-1}$. After reaching the set temperature the samples were held for 2 h at the target temperature and finally allowed to cool freely.

### 2.2. Research Techniques

For the imaging of the surface morphology and microstructure of as-deposited and additionally heat-treated Ni coatings, the focused ion beam/scanning electron microscopy (FIB/SEM) was used. Surface imaging and coating cross-sections were completed using SEM/Xe-PFIB FEI Helios microscope (FEI, Hillsboro, OR, USA). The image of a cross-section was registers in immersion mode of the microscope (2 kV accelerating voltage). The same mode was used for the imaging of coatings surface morphology.

To assess the changes in the coatings structure X-ray diffraction (XRD) analysis was performed using a Siemens D5000 diffractometer (Siemens AG, Munih, Germany) with CuK$\alpha_1$ radiation ($\lambda$ = 0.15406 nm) source. Measurements were made at 30° to 100° $2\theta$ angle ranges, 0.02° step and of 0.26° min$^{-1}$ scanning rate at ambient temperature. Metallic Ni from coating and Cu from base material were confirmed by the reference patterns from Powder Diffraction Files database (International Centre for Diffraction Data PDF-4 base). Scherrer formulae (Equation (1)) was used to calculate the crystallite size for the most intense peak for Ni with crystal plane (111):

$$D = \frac{K \cdot \lambda}{B \cdot \cos\theta} \tag{1}$$

The physical meaning of the symbols in Equation (1) is as follows: $D$—mean crystallite size in the direction perpendicular to (hkl) plane of the reflexes (nm), $\lambda$ the X-ray wavelength (0.154 nm), $K$—the constant for crystals in a cubic system (0.9), and $B$—the full width at half maximum (rad) of the peak at angle $\theta$ (rad) excluding hardware peak broadening:

$$B = \sqrt{(\beta_{FWHM}^2 - \beta_0^2)} \tag{2}$$

where $\beta_{FWHM}$—FWHM of Ni diffraction peak, $\beta$—instrumental broadening (in radian).

Microhardness of Ni coatings, as a function of heat treatment temperature, was measured using CSM Micro-Combi-Tester (CSM Instruments, Needham, MA, USA). The measurements were performed in accordance with ISO 14577-1:2015 using diamond Vickers indenter. It was observed that the measurement of microhardness from the coating surface was burdened with too large error resulting from its significant roughness. On the other hand, the possible influence of the oxidized nickel layer on the measured values was also taken into account. To avoid these inconveniences, a new series of coatings with the thickness about 20 μm has been deposited and the microhardness has been measured on the coatings cross-section. The loading and unloading rates equaled to 200 mN min$^{-1}$,

while the maximal force $F_{max}$ = 100 mN was applied for 10 s. These parameters ensured low indentation depths ($h_m$ = 720–830 nm) and ensured a negligible interaction of the indenter with the base material. The coatings were duplicated, and 20 indentations on each coating cross-section (with 100 μm spacing) were registered. Finally, according to Oliver and Pharr method [14], the indentation microhardness ($HV_{IT}$) and indentation modulus of elasticity ($E_{IT}$) were calculated.

The chemistry of coating's surface subjected to heat-treatment was analyzed by X-ray photoelectron spectroscopy (XPS) using a SPECS PHOIBOS 100 spectrometer (SPECS, Berlin, Germany) with Al source (1486.7 eV, non-monochromatized, 250 W for high resolution spectra). Ion etching during XPS analyses was carried out by $Ar^+$ sputtering with the beam energy 4 keV and a beam current density 7.5 μA $cm^{-2}$. Processing and fitting of the spectra were carried out in SPECLAB2 and CasaXPS v. 3.19 software, where a Gaussian–Lorenzian curve profile and a Shirley baseline were used. The energy 284.8 eV for the C 1s peak was used as the reference.

Electrochemical impedance spectroscopy (EIS) and linear polarization resistance (LPR) technique have been chosen to assess the influence of heat treatment of the coatings on their corrosion resistance. Non-deaerated 0.05 mol $dm^{-3}$ solution of NaCl was selected as the corrosive environment. Experience shows that the use of a less aggressive (than 0.5 mol $dm^{-3}$ NaCl) environment in combination with a longer exposure time of the material in this environment (168 h) can often deliver more information about the changes in corrosion process. The measurements were carried out in a 400 mL Autolab corrosion cell with Interface 1010E potentiostat (Gamry, Warminster, PA, USA). A classic three-electrode system was used: the working electrode (geometric area 1 $cm^2$), the counter electrode—stainless steel rod (geometric area 4 $cm^2$) and the Ag | AgCl (3 M KCl) reference electrode (Metrohm) mounted in a Luggin capillary. Impedance spectra were recorded at the open circuit potential ($E_{OC}$) during 168 h of exposure to NaCl solution with a resolution of 10 pts/dec., in a frequency range from 100 kHz to 1 mHz and 5 mV (rms) *ac* signal. LPR was performed by polarizing the samples from −15 to +15 mV vs. $E_{OC}$ with a scan rate of 1 mV $s^{-1}$. The system was locked in a Faraday cage to avoid interference from an external electric field. Analysis of impedance spectra and determination of polarization resistance were performed using ZView® (Scribner Associates, Southern Pines, NC, USA) and EchemAnalyst (Gamry) software.

## 3. Results and Discussion

### 3.1. The Effect of Heat Treatment on the Structure and Microhardness of the Coatings

Structure of 7 μm thick "as-deposited" nickel coating was analyzed by XRD. The diffractogram (Figure 1) shows split peaks at 2θ about 43.3°, 50.4°, 74.1° and 89.9° that belong to Cu base material (ICDD Card No. 00-004-0836). Diffraction peaks visible at 2θ: 44.6°, 51.9°, 76.7° and 92.8° are characteristic of Ni (111), (200), (220) and (311) crystal planes, respectively, (ICDD Card No. 00-004-0850) for the face centered cubic (FCC) structure of metallic nickel.

The calculated crystallite size for the coatings depending on the annealing temperature is presented in Table 1. The calculated *D* values and not very intense and broad Ni peaks at the diffractograms confirmed nano-crystallinity of all tested coatings. Abbott et al. [15] have also reported similar structure of Ni electrodeposited from ethaline.

According to the literature, the post-deposition heat treatment and subsequent increase of crystallite size should affect coatings corrosion resistance and microhardness as indicated by Laszczyńska et al. [16]. Initially, for the as-deposited Ni coating, the indentation microhardness amounted to 589 Vickers (Table 1) which was 18–38% higher than those (425–500 HV) measured by Danilov et al. [17] for similar Ni coatings deposited in DES and 5–14% higher than these measured by Abraham et al. [18] for coatings from conventional Watt's plating baths (517–560 HV). After the heat treatment at 100 °C for 2 h the microhardness has raised to ca. 670 Vickers. The highest microhardness has been observed after heat treatment at 200 °C (803 Vickers). Further increase in the heating temperature

up to 400 °C was not favorable, because a decrease in microhardness to 475 Vickers was observed (Table 1). The indentational modulus of elasticity ($E_{IT}$) of the deposited nickel did not change after heat treatment at 100 °C and remained at 183–186 MPa. Only after heat treatment at 200 °C $E_{IT}$ it started to increase and it reached a maximum (237 MPa) at 300 °C (Table 1). An increase in the heat treatment temperature (up to 400 °C) of the nickel coating caused an increase in the size of the crystallites (Table 1). Increase in the size of the crystallites is often associated with an increase in microhardness. In the current work, an increase in the microhardness of nickel coatings was found with the increase of the heat treatment temperature, but only in the range up to 200 °C. During annealing at 300 °C and 400 °C, there was a significant reduction in microhardness, even though the size of the crystallites increased continuously. It follows that the heat treatment temperature is not the only factor influencing microhardness.

**Table 1.** The estimated crystallite size (*D*), indentation microhardness ($HV_{IT}$) and indentation modulus of elasticity ($E_{IT}$) of Ni coatings depending on the heat treatment temperature.

| Parameter | As-Deposited | 100 °C | 200 °C | 300 °C | 400 °C |
|:---:|:---:|:---:|:---:|:---:|:---:|
| *D* (nm) | 9.8 | 12.7 | 23.5 | 31.2 | 35.2 |
| $HV_{IT}$ | 588.7 (29.7) | 666.5 (45.2) | 803.4 (72.1) | 618.3 (31.9) | 475.4 (50.7) |
| $E_{IT}$ (MPa) | 185.6 (20.8) | 183.3 (20.3) | 211.0 (33.9) | 237.3 (34.9) | 213.1 (30.5) |

values in brackets are the standard deviation for microhardness and the modulus of elasticity.

To verify that heat-treated nickel coatings agree with Hall–Petch relation, the relationship between microhardness and inverse of square root of grain size has been presented in Figure 2. It was found that Ni coatings followed Hall–Petch relationship up to a grain size of ca. 24 nm, that corresponds to the post-deposition heat treatment temperature of 200 °C. Further decrease in grain size (as-deposited coating and coating annealed at 100 °C) caused a decrease in microhardness thereby signifying the breakdown of the Hall–Petch relationship. Schuh and Nieh [19] have reported that Hall–Petch inflection occurs for pure Ni with crystallite size of 12–15 nm. The differences in values of this inflection may be due to the method of obtaining the coating. Among the deformation mechanisms related to Hall–Petch breakdown, researchers pay the greatest attention to the following two: grain boundary sliding and Coble creep. Other mechanisms indicated by Schuh and Nieh [19] and Naik and Walley [20] that should be taken into account as responsible for the decrease in the strength of nanometric materials are grain boundary diffusion, amorphous limit and grain rotation. Abraham et al. [18] also indicated that the microhardness of nickel coatings is determined by percolation of large grains and increase in the volume friction of the abnormal grains.

### 3.2. The Effect of Heat Treatment on the Morphology and Microstructure of the Coatings

High-resolution scanning electron microscopy allowed to assess the coatings almost at a nano-scale and revealed interesting surface morphology (Figure 3). They were characterized by the developed surface, especially for the as-deposited one and these heat-treated at 100 and 200 °C. The morphology of coatings treated at these temperatures was quite interesting: nano-sized microplates embedded in granular agglomerates (Figure 3a–c). After heat treatment at 300 °C the morphology of the coatings began to change (Figure 3d) and after heat treatment at 400 °C it no longer resembled that at 25 °C (Figure 3e). The nano-flakes observed previously were almost completely embedded in the matrix of oxidized nickel (Figure 3e).

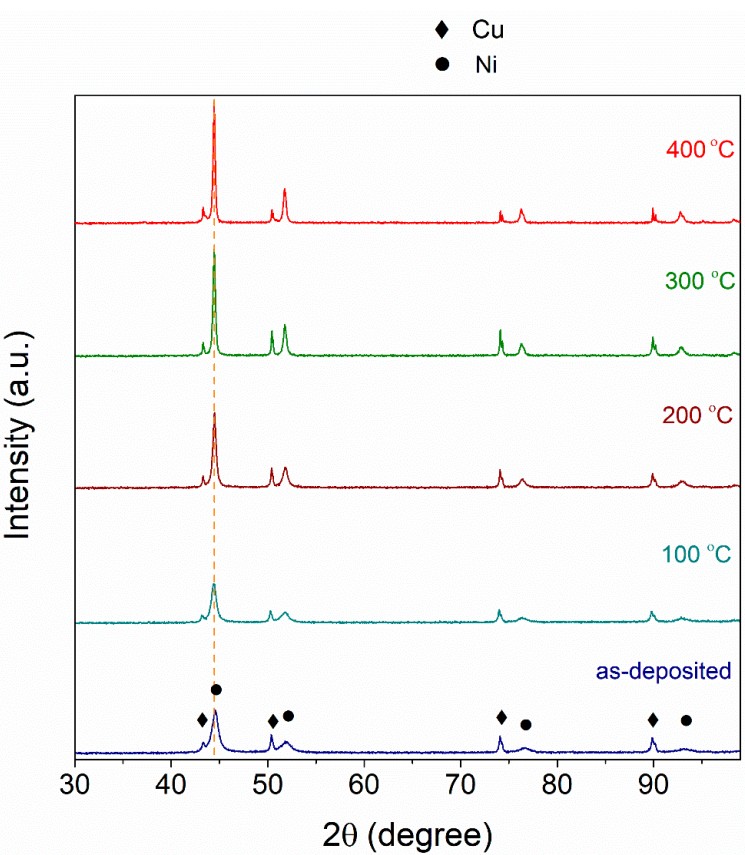

**Figure 1.** X-ray diffractograms for Ni coatings (as-deposited coating and 100 °C–400 °C—heat treated temperature) obtained in a plating bath composed of ChCl and EG (1:2 molar ratio) and 1 mol dm$^{-3}$ NiCl$_2\cdot$6H$_2$O. Plating parameters: $j_c$ = 6 mA cm$^{-2}$, $t$ = 90 min, $T$ = 70 °C. Coating thickness 7–8 μm.

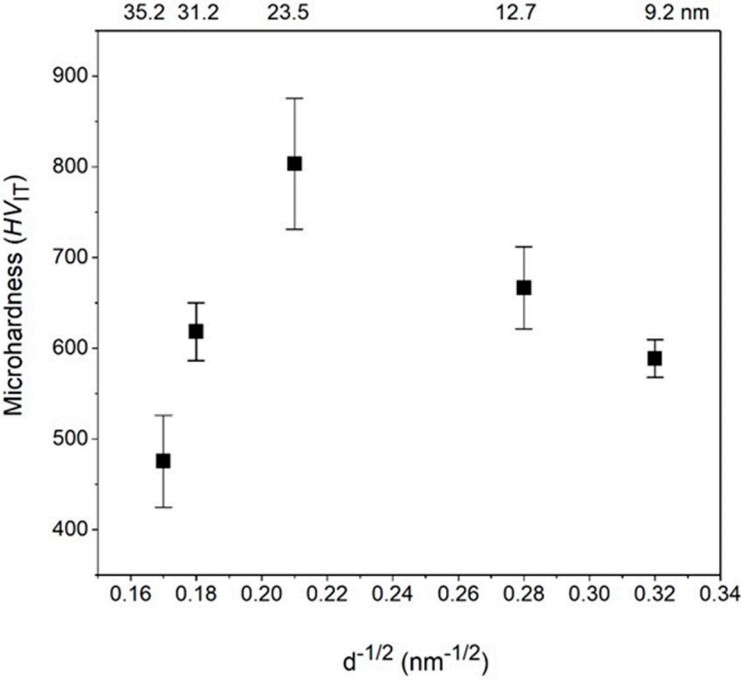

**Figure 2.** Hall–Petch plot of as-deposited Ni and heat-treated Ni coatings obtained in a plating bath composed of ChCl and EG (1:2 molar ratio). Mean crystallite size (in nm) is given on the top X axis.



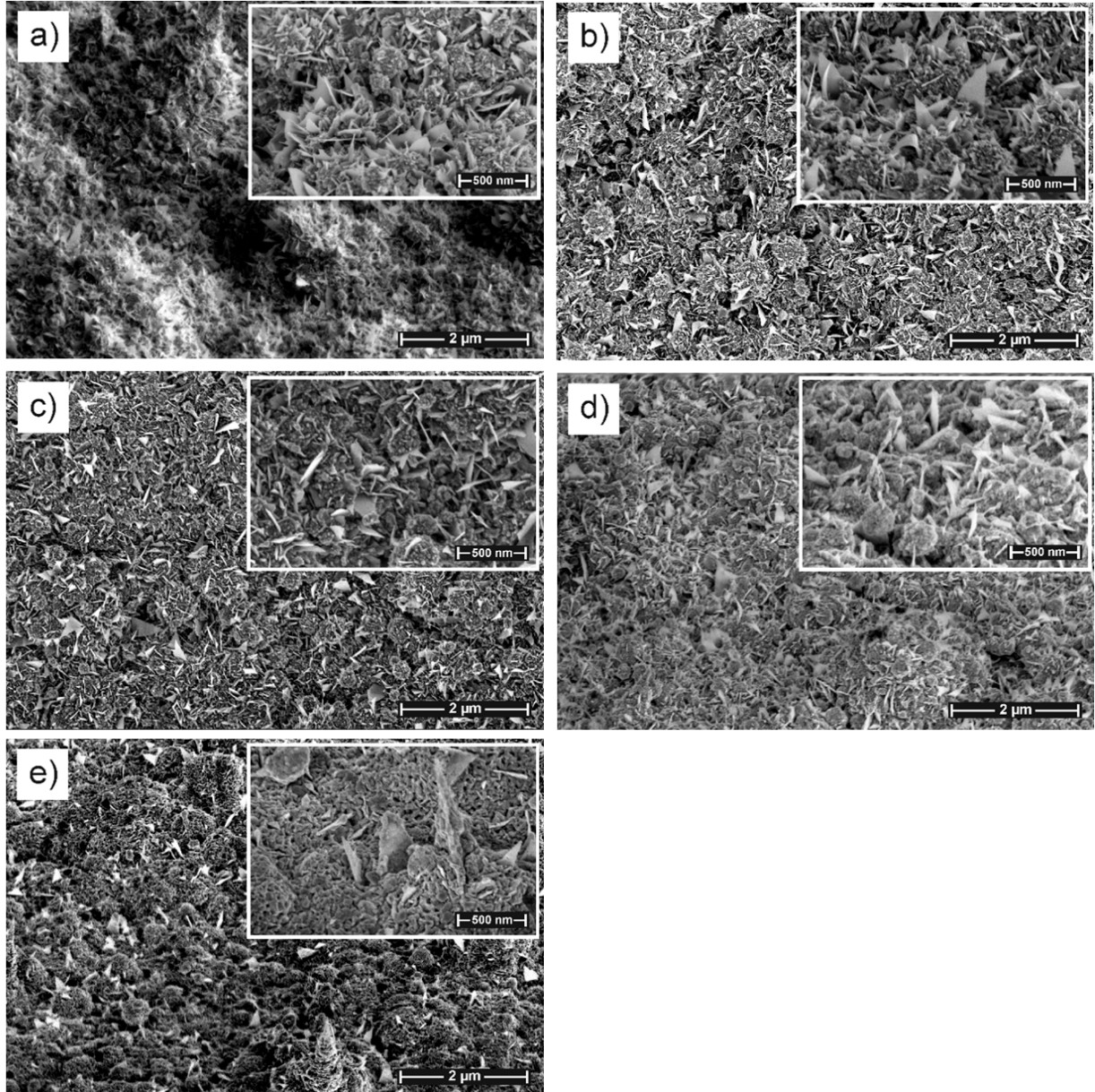

**Figure 3.** Morphology of Ni coatings deposited from ChCl: ethylene glycol bath with 1 mol dm$^{-3}$ NiCl$_2 \cdot$6H$_2$O at 6 mA cm$^{-2}$ for 90 min at 70 °C before (**a**) and after 2 h heat-treatment at: 100 (**b**), 200 (**c**), 300 (**d**) and 400 °C (**e**).

To better understand the effect of annealing of the coatings on their properties, microscopic analyzes were performed using the SEM/PFIB double beam microscope. Therefore, two samples have been chosen: as-deposited and annealed at 300 °C—for the latter a decrease of microhardness and visible change in surface morphology have been observed. The cross-sections presented in Figure 4 confirmed the mean coating thickness of 7–8 μm that corresponds to deposition rate ~5 μm h$^{-1}$. The microstructure of both coatings was homogenous, and free of cracks (Figure 4a,c).

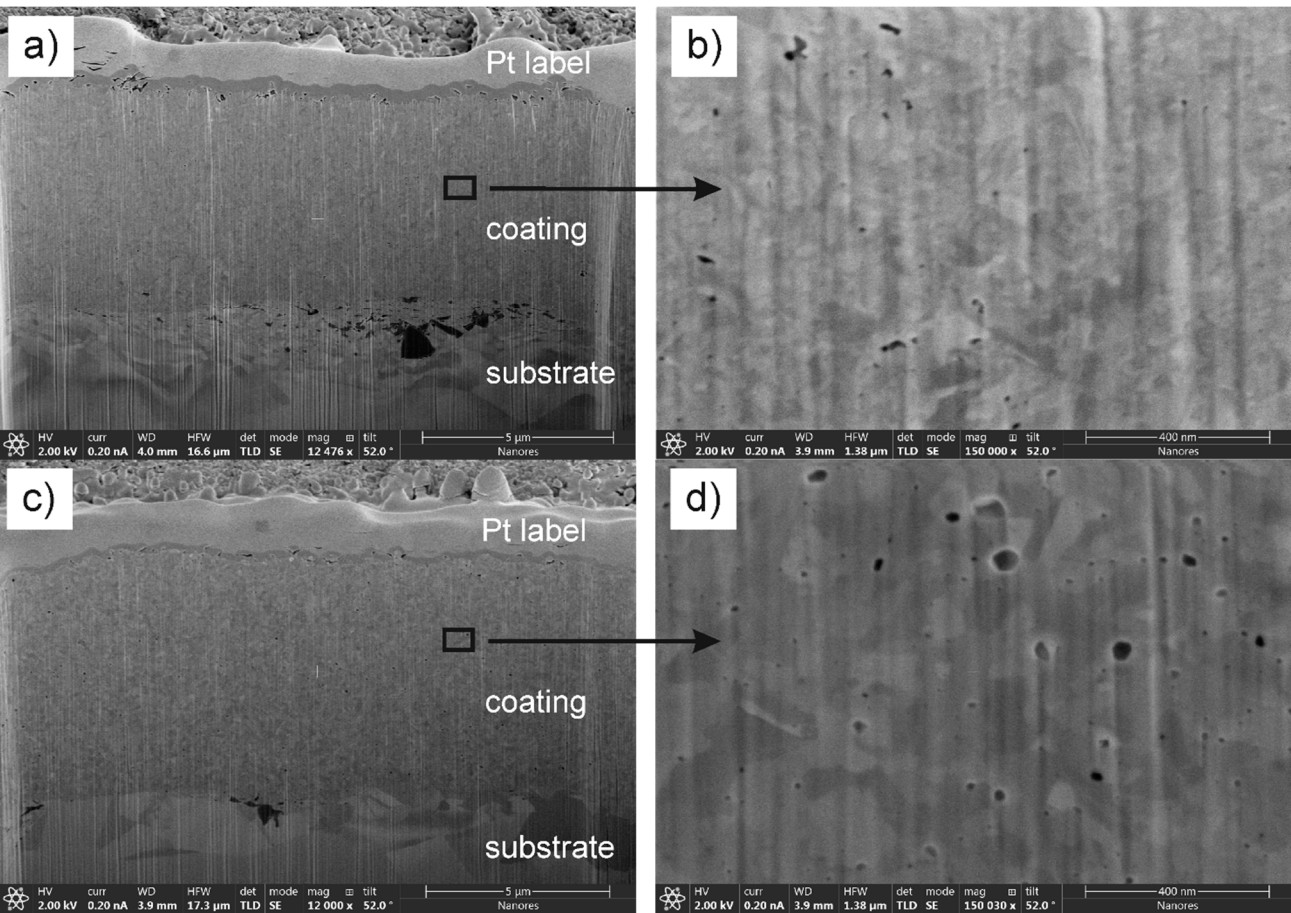

**Figure 4.** Coatings cross-sections demonstrating the microstructure at a micro- and nano-scale of Ni coatings before (**a**,**b**) and after (**c**,**d**) heat treatment at 300 °C for 2 h, deposited from ChCl: ethylene glycol DES containing 1 mol dm$^{-3}$ NiCl$_2$·6H$_2$O (at 6 mA cm$^{-2}$ for 90 min at 70 °C).

Microscopic observations carried out at high magnifications provided new information on the microstructure of both nickel coatings (as-deposited and heat-treated at 300 °C). First of all, microphotographs in Figure 4 clearly show that the as-deposited Ni coating was composed of fine agglomerates of crystallites. In the case of coating annealed at 300 °C, the dimensions of these agglomerates often exceeded 200 nm. Secondly, in the microstructure of the latter, a lot of voids with oval shape and size much larger than in the case of as-deposited Ni coating were observed (Figure 4d). The size of the voids ranged from single to 20 nm for the as-deposited Ni coating, while the coating annealed at 300 °C was characterized by voids even up to 80 nm in diameter. This may indicate at least two situations: (1) During the electrodeposition, some residues of organic bath (or partly reduced Ni compounds) were embedded in the structure of the Ni metallic matrix and then thermally decomposed leaving free spaces or (2) During annealing microstructural changes in the coating occur, which resulted in the increase of the observed voids.

The porosity of Ni coating treated at 300 °C for 2 h observed using the SEM/PFIB technique (Figure 4d) may be the reason why this coating exhibited lower microhardness (Table 1). In this situation, the hypothesis about the possible decomposition of some residues of organic compounds, originated from plating bath and incorporated during the electrodeposition of Ni coating as a result of heat-treatment at 300 and 400 °C could not be excluded.

### 3.3. The Effect of Heat Treatment on the Surface Chemistry: XPS Analysis

Table 2 presents the composition of the surface of as-deposited nickel coating, and the coatings annealed for 2 h at: 100, 200, 300 and 400 °C. To better visualize the morphology of the oxidized nickel layers the surface compositions have been presented both for the "as-received" and for the sputter cleaned (Ar$^+$ beam for 1 min at 1.5 keV and 2 μA cm$^{-2}$) surface states.

**Table 2.** Changes in chemical composition (at.%) of the outer surface of the as-deposited Ni coating and the Ni coatings after annealing for 2 h at: 100, 200, 300 and 400 °C. The surface of nickel coatings was analyzed immediately after deposition plus annealing and after a gentle/mild Ar+ cleaning (1.5 keV, 2 μA cm$^{-2}$, 60 or 90 s).

| | Coating | Ni 2p | O 1s | C 1s | O:Ni |
|---|---|---|---|---|---|
| fresh | "as received" immediately after deposition, rinsing and drying | 44.79 | 18.68 | 36.53 | 0.42 |
| | +Ar$^+$ 60 s | 69.03 | 15.30 | 15.67 | 0.22 |
| 20 °C | "as received" after 24 h of exposure to air at 20 °C | 38.32 | 30.87 | 30.82 | 0.81 |
| | +Ar$^+$ 60 s | 62.93 | 19.49 | 17.58 | 0.31 |
| 100 °C | "as received" after 2 h annealing at 100 °C | 29.19 | 26.46 | 44.35 | 0.91 |
| | +Ar$^+$ 60 s | 49.32 | 20.27 | 30.41 | 0.41 |
| 200 °C | "as received" after 2 h annealing at 200 °C | 26.93 | 40.26 | 32.80 | 1.49 |
| | +Ar$^+$ 90 s | 40.63 | 39.61 | 19.76 | 0.97 |
| 300 °C | "as received" after 2 h annealing at 300 °C | 32.66 | 50.67 | 16.67 | 1.55 |
| | +Ar$^+$ 60 s | 43.42 | 51.28 | 5.3 | 1.18 |
| 400 °C | "as received" after 2 h annealing at 400 °C | 25.4 (+16.0 Cu) | 44.52 | 14.11 | 1.75 |

The presentation of the chemical composition of the nickel coating (Table 2) immediately after deposition plus annealing and after sputter cleaning with Ar$^+$ ions (at two different points in the layer depth profile) shows the influence of the contact of the surface of the samples with the atmospheric environment. Freshly deposited Ni coating (denoted as "as-deposited") contained approx. 36 at.% carbon and its compounds (bonds) with oxygen (Table 2). The deconvolution of the C 1s spectrum showed the bond distribution typical for the outer layers: C-C (C-H) 284.8 eV (reference energy), COH/COR 286.3, C=O 287.7 and COOH/CO$_3$ > 288 eV (it is clearly visible in Figure 5). An abrupt decrease in the carbon content (up to 15 at.%—please refer to Table 2) after gentle sputter cleaning with Ar$^+$ ion beam of the top coating proved the small thickness of the carbon contamination layer.

According to the XPS survey spectra (not presented here), the dominant component of "as-deposited" coating was nickel—44.8 at.% (Table 2). The analysis of particular oxidation states of nickel was performed on the basis of Ni 2p spectral regions and the chemical shifts. Ni 2p spectra together with deconvolutions for the obtained Ni coating, and the nickel coatings annealed for 2 h at: 100, 200, 300 and 400 °C, are presented in Figure 6. The main assumption during deconvolution was the presence of: Ni(0), Ni(II) and Ni(III) in the form of NiO, Ni(OH)$_2$ and NiOOH. The whole complex deconvolution procedure has already been presented in earlier works of Winiarski et al. [1].

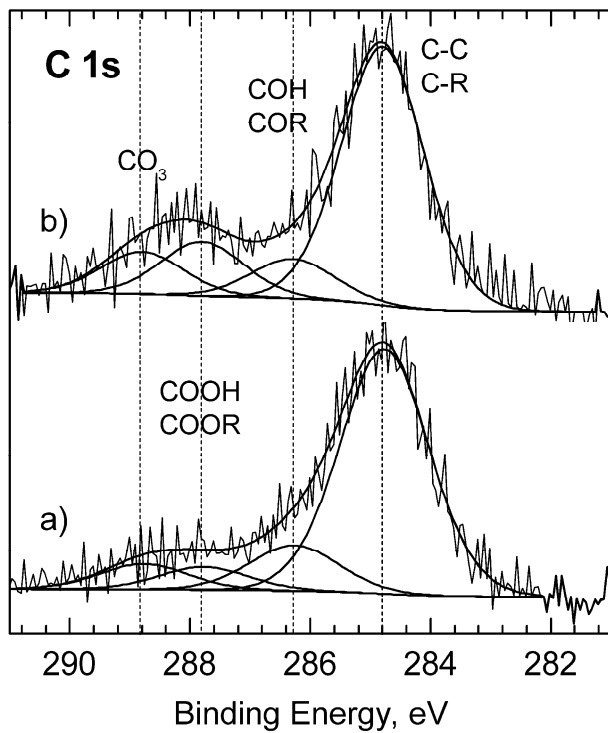

**Figure 5.** XPS C 1s spectra with deconvolution for Ni coatings: (**a**) "as-deposited", (**b**) "as-deposited" and stored open to air for 24 h, analyzed in the "as-received" form.

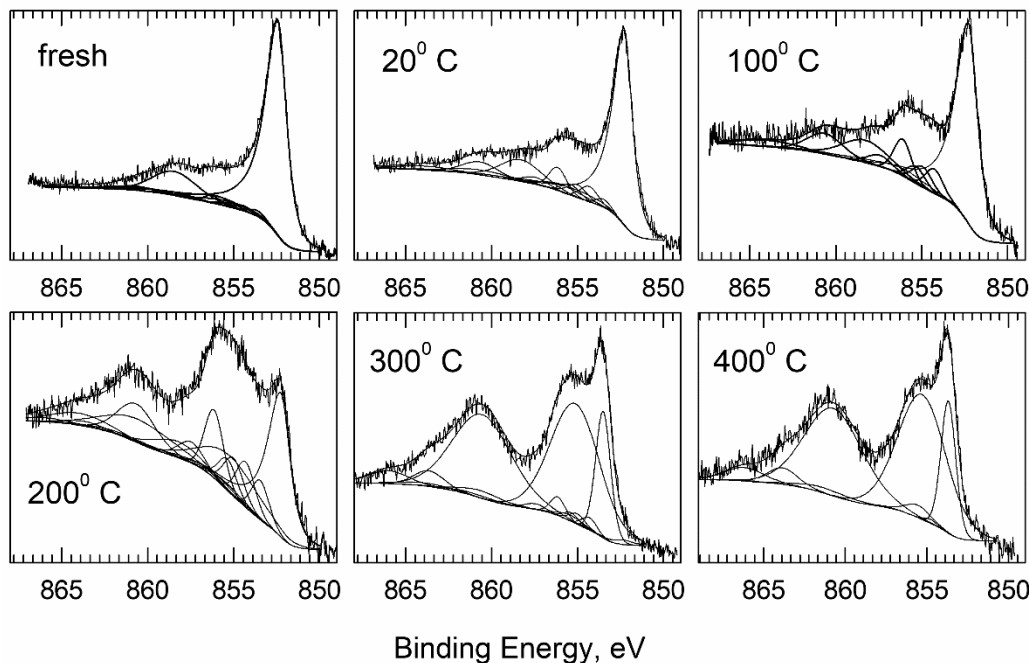

**Figure 6.** XPS Ni $2p_{3/2}$ spectra of the samples with deconvolution indicating the components: Ni(0), NiO, NiOOH and Ni(OH)$_2$.

As shown by the deconvolution of Ni $2p_{3/2}$ spectra (Figure 6), metallic nickel was covered with its natural oxides. The ratio of oxidized forms of nickel to metallic nickel (Ni$_{ox}$:Ni$_{metal}$) on the analyzed surface in the "as-deposited" form was 0.16 (Table 3). The top layer of the coating additionally contained significant amounts of oxygen bound to the "natural" carbon (the ratio of total oxygen to nickel was O:Ni = 0.42—please refer to Table 2). Conditioning the sample at room temperature for 24 h, increased the total surface

oxidation state to O:Ni = 0.81 (Table 2). The increase in the O:Ni ratio was the result of the increasing share of nickel oxides (expressed in the $Ni_{ox}$: $Ni_{metal}$ ratio) from 0.16 to 0.42 after 24 h of conditioning (Table 3). The amount of surface carbon and its compounds (bonds) with oxygen remained at the same level. As expected, annealing of the samples with nickel coatings for 2 h at temperatures of 100, 200, 300 and 400 °C deepened the process of oxidation of their surface (Table 3). Another feature was the increase in the proportion of NiO from 2.4 to 30% in relation to $Ni_{total}$ after increasing the annealing temperature from 100 to 200 °C (Table 3). Undoubtedly, this observed progressive phase transition was reflected in the corrosion resistance of Ni coatings annealed at 200, 300 and 400 °C.

**Table 3.** Shares of oxidized forms of Ni (in %) detected on the surface of nickel coatings depending on their post-deposition heat treatment temperature. The compositions listed above concern the surface analyzed in the "as received" state (i.e., without $Ar^+$ sputter clearing).

|  | $Ni_{metal}$ | $Ni(OH)_2$ | NiO | NiOOH | $Ni_{ox}$:$Ni_{metal}$ |
|---|---|---|---|---|---|
| as-deposited | 86.10 | 0.64 | 6.97 | 6.29 | 0.16 |
| as-deposited (stored for 24 h, open to air) | 70.44 | 0.04 | 6.38 | 23.14 | 0.42 |
| 100 °C, 2 h | 66.99 | 0.76 | 2.38 | 29.86 | 0.49 |
| 200 °C, 2 h | 31.02 | 2.09 | 29.56 | 37.33 | 2.22 |
| 300 °C, 2 h | 1.44 | 1.91 | 87.64 | 9.00 | 68.44 |
| 400 °C, 2 h | 0.18 | 4.56 | 95.25 | 0.00 | >500 |

After annealing of the coating at 400 °C, additionally, significant amount of Cu was determined (16 at.%, Table 2). Copper was the substrate for the deposited Ni coatings and most likely it segregated onto the coating surface during heat treatment along the $Ni/Ni_{ox}$ grain boundaries or through discontinuities in the coating

### 3.4. The Effect of Heat Treatment on Corrosion Resistance of the Coatings

#### 3.4.1. Linear Polarization Resistance

First analysis of the samples was performed by a non-destructive LPR technique, which allows to estimate the instantaneous corrosion rate very quickly. The resulting trend of polarization resistance value ($R_p$) measured during exposure of the samples in 0.05 mol $dm^{-3}$ NaCl solution is presented in Figure 7.

Heat treatment of nickel coatings caused a significant increase in their corrosion resistance (higher $R_p$ values). After annealing at 100 °C, the polarization resistance measured during the entire 168-h exposure period reached values several times higher (>0.8 M$\Omega$ $cm^2$) than those for as-deposited Ni coating (~0.1 M$\Omega$ $cm^2$). A further increase in corrosion resistance ($R_p$ > 2 M$\Omega$ $cm^2$) was observed after raising temperature to 200 °C (Figure 7). One possible reason for the increase in $R_p$ is the formation on the coatings surface was a relatively compact layer consisting mainly of NiOOH and NiO. Another reason may be the reduction of active area on the coatings' surface during heat treatment, as reported by Laszczyńska et al. [4] and Popczyk et al. [21]. At 300 °C, however, the corrosion resistance decreased. Further heat treatment at 400 °C caused that the change in polarization resistance (both in terms of trends and the values) became similar to that of as-deposited Ni. What is interesting, is that the corrosion resistance of the obtained nickel coatings varied depending on the heat treatment temperature in a manner similar to the microhardness (Table 1). Microhardness (Table 1) reached its maximum for coating annealed at 200 °C. These results are consistent with the SEM analysis (Figure 3), which showed a clear transition in the morphology of the heat-treated coating above that temperature. It can be assumed that the increase in the porosity of the coatings (observed in Figure 4c,d) translated into the deterioration of its anti-corrosion properties.

During the LPR measurements, attention was also paid to monitor the corrosion potential ($E_{corr}$) of the coatings in NaCl solution (Figure 8). Although it is not a parameter that determines corrosion resistance, it gives some information about the state of the surface

of the coating and its interaction with the corrosive environment. It was noticed that the corrosion potentials of the coatings after heat treatment were higher (by more than 300 mV) than $E_{corr}$ of the as-deposited Ni coating. There are several possible reasons for the increase in $E_{corr}$. One of them is to create, on the Ni coating, a compact semiconductive layer consisting of NiOOH and NiO. After heat treatment at 400 °C, a slight decrease in $E_{corr}$ was observed. The higher porosity of the NiO layer was most likely the cause. Moreover, on the surface of the coating annealed at 400 °C, 16 at.% of Cu was detected in the XPS analysis. This very large (on a nano-scale) amount of copper, which segregated to the surface of the sample, should probably increase the $E_{corr}$. Oriňáková et al. [22] have observed similar phenomenon—a significant change in $E_{corr}$ of Ni coatings deposited on a Fe base material due to iron diffusion through nickel matrix during heat treatment.

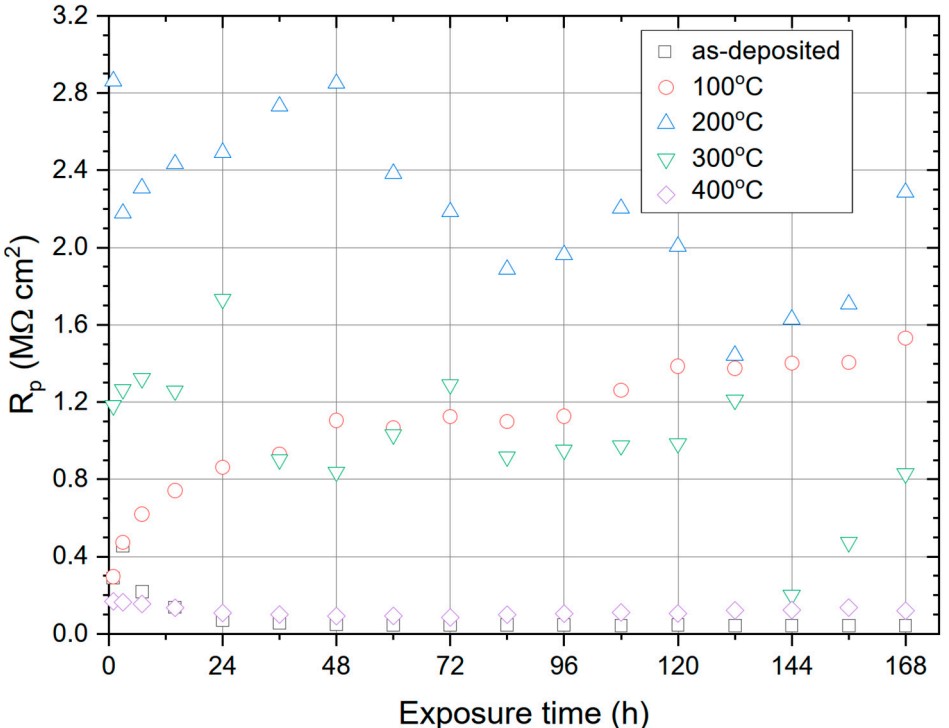

**Figure 7.** Polarization resistance measured in 0.05 mol dm$^{-3}$ solution of NaCl during 168 h of exposure of as-deposited Ni and heat-treated Ni coatings obtained in a plating bath composed of ChCl and EG (1:2 molar ratio) and 1 mol dm$^{-3}$ NiCl$_2$·6H$_2$O (current density $j_c$ = 6 mA cm$^{-2}$, 90 min, 70 °C, coating thickness 7–8 μm).

### 3.4.2. Electrochemical Impedance Spectroscopy

EIS technique was used to assess the interaction of temperature-modified coating surface with the corrosive environment. Impedance spectra were collected in the potentiostatic mode after 7 days of exposure of the coatings in non-deaerated 0.05 mol dm$^{-3}$ NaCl solution. Nyquist representation of EIS spectra (Supplementary Material Figure S1) indicates a significant increase in the corrosion resistance of coatings after subjecting them to the heat-treatment.

The spectra presented in Figure 9 in the Bode notation provided much more information. The coarse analysis of the spectrum for the as-deposited Ni coating revealed the impedance modulus at 0.001 Hz ($|Z|_{0.001Hz}$) equal to ca. 40–50 kΩ cm$^2$ (Figure 9a) and two not clearly separated time constants (at ca. 10 Hz and 0.1 Hz) in the phase angle ($\theta$) frequency dependence (Figure 9b). Heating the coating at 100 °C caused disappearance or overlapping of these two time constants and finally only one broad peak remained visible at about 0.2 Hz in the phase angle ($\theta$) spectrum (Figure 9b). The very large increase in the modulus of impedance was also noteworthy—from 40 to 50 kΩ cm$^2$ for as-deposited Ni

coating to about 1 MΩ cm$^2$ for the annealed one (100 °C)—Figure 9a. This trend also continued for the coating heat treated at 200 °C after 168 h of its immersion in 0.05 mol dm$^{-3}$ NaCl, i.e., the measured impedance was even greater (Figure 9a). Looking at the phase angle spectrum, no difference was noticed between that for 100 and 200 °C (Figure 9b). A certain noticeable change in the shape of the impedance spectra appeared after heat treatment of nickel coating at 300 °C. Again, as for the as-deposited coating, there were probably two poorly visible phase angle maxima, which could indicate the existence of two time constants (Figure 9b). However, only a minor decrease in the impedance modulus ($|Z|_{0.001Hz}$) has been observed (Figure 9a). A temperature rise to 400 °C caused an almost 10-fold decrease in the impedance modulus $|Z|_{0.001Hz}$ value, from ca. 1 MΩ cm$^2$ to just over 100 kΩ cm$^2$ (Figure 9a) which is equivalent to a significant reduction in corrosion resistance of the coating material. Another characteristic feature was the presence of two phase angle maxima for this coating (Figure 9b).

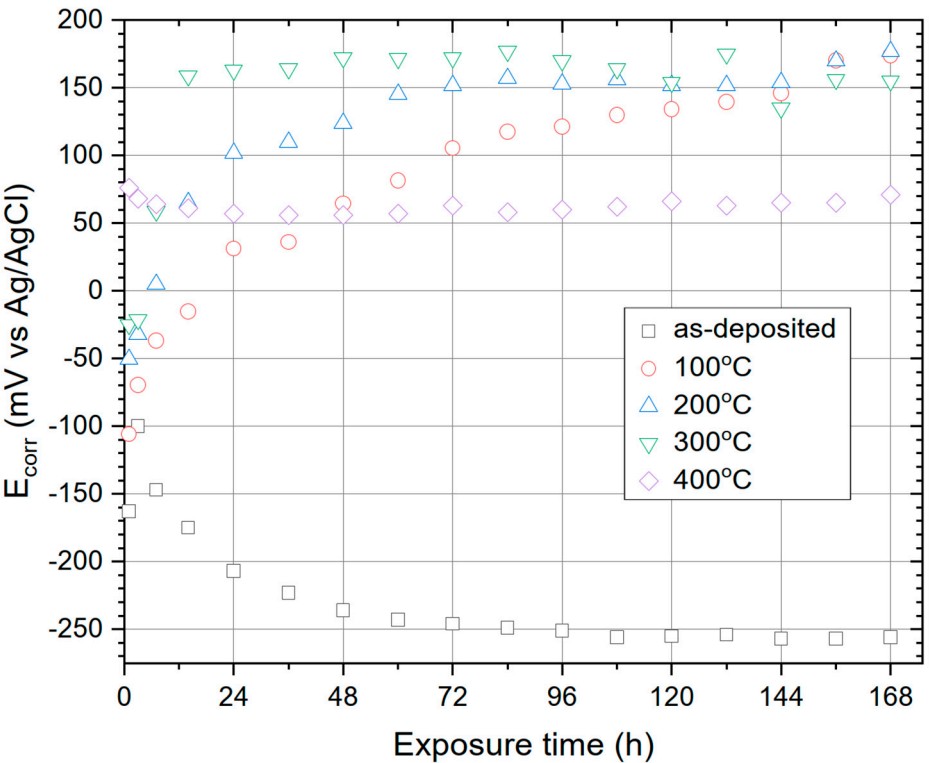

**Figure 8.** Corrosion potential measured in 0.05 mol dm$^{-3}$ NaCl solution during 168 h of exposure of Ni coatings obtained in a plating bath composed of ChCl and EG (1:2 molar ratio) and 1 mol dm$^{-3}$ NiCl$_2$·6H$_2$O ($j_c$ = 6 mA cm$^{-2}$, 90 min, 70 °C, coating thickness 7–8 μm).

A cursory analysis of the Bode plot in Figure 9 indicated the use of two electric equivalent circuits (EEC's): a single time constant model ("model I" in Figure 10) for the coatings annealed at 100 and 200 °C, and a double time constant model ("model II" in Figure 10) for Ni coatings "as-deposited" and annealed at 300 and 400 °C. These models use a constant phase element (CPE) in order to fit better to the experimental spectra. The impedance of the CPE can be expressed with Equation (3), where: $Y_0$ is a time constant parameter (Ω$^{-1}$ cm$^{-2}$ s$^\alpha$), $\omega$ is the angular frequency of the *ac* signal and $\alpha$ is the exponent of CPE.

$$Z_{CPE} = Y_0^{-1}(j\omega)^{-\alpha} \tag{3}$$

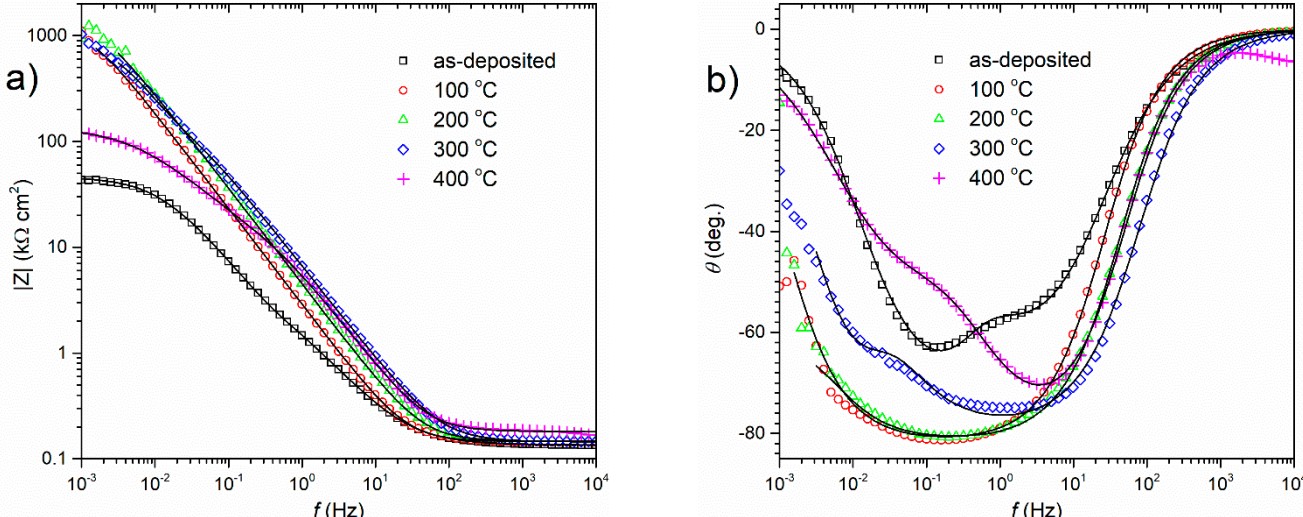

**Figure 9.** Bode plot of impedance spectra recorded after 168 h exposure in 0.05 mol dm$^{-3}$ NaCl for the as-deposited and heat-treated nickel coatings for 2 h at: 100; 200; 300 and 400 °C. Frequency dependence of the impedance modulus (**a**) and the phase angle (**b**).

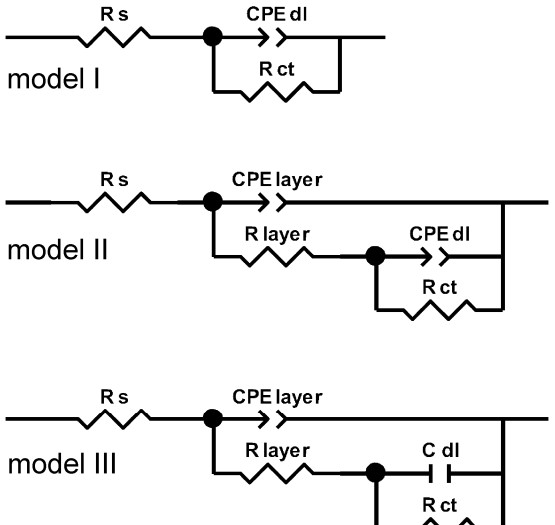

**Figure 10.** Electric equivalent circuits for the corrosion of Ni coatings.

The physical meaning of electric elements used in models I and II (Figure 10) was as follows: $R_s$—the resistance of sodium chloride solution, $R_{ct}$—the charge transfer resistance connected with metal oxidation, $Y_{0,dl}$ and $\alpha$—the double layer capacitance ($C_{dl}$). Model II (Figure 10) has been expanded by an additional CPE element ($Y_{0,layer}$ and $\alpha$—correspond to the electrical capacitance of the surface oxidized layer) connected in parallel with a resistor ($R_{layer}$—the resistance associated with this layer). Both models ensured a good fit to the experimental impedance spectra, yielded in low $\chi^2$ (~$10^{-4}$) and residual errors: 0.1–6%.

Randles circuit (model I in Figure 10) is usually a good starting point for any consideration. However, the use of a model that would simulate not only the Faradaic process, but also the properties of the oxidized layer on heat-treated nickel coating (the presence of which was evidently demonstrated by previous spectroscopic and microscopic analyzes) turned out to be more favorable for consistent interpretation of all the results presented in this work. Ultimately it was decided to use model II, but with a minor modification—model III in Figure 10. In this model it was possible to use a capacitor instead of CPE to describe the double layer capacitance ($C_{dl}$). This choice was dictated by the literature research on passive layers for stainless steels, nickel, titanium and aluminum using the EIS technique.

The results of the fitting are presented in Table 4. Only for the coating annealed at 400 °C the calculation of a double layer capacitance ($C_{dl}$) required a different approach. It turned out that only in the case of this coating it was necessary to use the CPE element, as the pure capacitor did not allow for a good fit. The effective capacitance, $C_{dl,surface}$ (F cm$^{-2}$), was calculated according to Hirschorn et al. [23] from Equation (4):

$$C_{dl,surface} = Y_0^{1/\alpha} \left( \frac{R_e R_t}{R_e + R_t} \right)^{(1-\alpha)/\alpha} \tag{4}$$

where $Y_0$—has meaning of the capacitance ($\Omega^{-1}$ cm$^{-2}$ s$^\alpha$); $\alpha$—is the exponent of CPE; $R_e$—is the Ohmic resistance; $R_t$—the parallel resistance. Finally, value of this capacitance has been presented in Table 4 for the "400 °C" sample.

**Table 4.** Fitting results for impedance spectra recorded after 168 h immersion in 0.05 mol dm$^{-3}$ NaCl solution of nickel coatings before and after post-deposition heat treatment. Model III from Figure 10 was used in the calculations.

| Sample | $R_s$ ($\Omega$ cm$^2$) | $CPE1$-$T$ ($\Omega^{-1}$ cm$^{-2}$ s$^{-P}$) | $CPE1$-$P$ | $R_{layer}$ (k$\Omega$ cm$^2$) | $C_{dl}$ ($\mu$F cm$^{-2}$) | $R_{ct}$ (k$\Omega$ cm$^2$) |
|---|---|---|---|---|---|---|
| as-deposited | 135 | $1.4 \times 10^{-4}$ | 0.78 | 6.0 | 62 | 40.5 |
| 100 °C | 145 | $6.5 \times 10^{-5}$ | 0.91 | 1524 | 69 | 846 |
| 200 °C | 145 | $4.1 \times 10^{-5}$ | 0.90 | 1740 | 90 | 895 |
| 300 °C | 144 | $3.0 \times 10^{-5}$ | 0.88 | 278 | 32 | 650 |
| 400 °C | 180 | $3.2 \times 10^{-5}$ | 0.90 | 18 | 6.5 * | 118 |

* for the "400 °C" sample, model II gave the best fit and therefore the effective capacitance has been calculated from CPE parameters according to the surface time-constant distribution proposed by Hirschorn et al. [23].

According to Table 4, the resistance connected with the barrier properties of the oxidized surface layer ($R_{layer}$) has very strong increased after post-deposition heat treatment at 100 °C from ca. 6 k$\Omega$ cm$^2$ to ca. 1.5 M$\Omega$ cm$^2$. This was accompanied by a twentyfold increase in charge transfer resistance ($R_{ct}$). Increasing the temperature to 200 °C resulted in a further, although not so spectacular, increase in the above-mentioned values of circuit parameters (Table 4). This is not surprising, as the LPR measurements performed first showed that the polarization resistance significantly exceeded 1.5 M$\Omega$ cm$^2$ (Figure 7). What undoubtedly attracted the most attention is the fact that the corrosion resistance of this coating collapsed after heat treatment at 300 °C. Charge transfer resistance decreased by more than 200 k$\Omega$ cm$^2$, but $R_{layer}$ decreased as much as six times. Finally, at the highest of the temperatures used (400 °C), $R_{layer}$ and $R_{ct}$ reached 18 and 118 k$\Omega$ cm$^2$, respectively, which was close to those calculated for non-modified Ni coating (Table 4). Undoubtedly, the deterioration of corrosion resistance observed during *ac* and *dc* polarization measurements of nickel coating after annealing at 300 °C is the result of changes in the surface morphology and surface chemistry. After the heat treatment at 200 °C, a large proportion of NiOOH and Ni(OH)$_2$ compounds in relation to NiO was observed, which probably ensures good barrier properties of surface layer. After annealing at 300 °C, nickel oxide was already dominant, and was this compound that builds the oxidized layer. It is also probable that the NiOOH formed during oxidation undergone a transformation into NiO at 300 °C. The consequence of these changes is the formation of the structure and morphology of the oxidized layer with much worse barrier properties. A slightly similar to $R_{ct}$ trend was observed for a double layer capacitance ($C_{dl}$). This capacitance was initially 62 $\mu$F cm$^{-2}$, slightly increased to 69 $\mu$F cm$^{-2}$ at 100 °C and 90 $\mu$F cm$^{-2}$ at 200 °C. After exceeding 300 °C, it dropped to 32 $\mu$F cm$^{-2}$, and at 400 °C to 6 $\mu$F cm$^{-2}$.

## 4. Conclusions

- Nanocrystalline nickel coatings were deposited on a copper base material from deep eutectic solvent of choline chloride and ethylene glycol (1:2 molar ratio) containing 1 mol dm$^{-3}$ NiCl$_2 \cdot$6H$_2$O under galvanostatic conditions at j$_c$ = 6 mA cm$^{-2}$ and the

bath temperature of 70 °C. The surface of 7–8 μm thick coatings were composed of spheroidal agglomerates with the size of several hundred nanometers interspersed with lamellar crystals.

- Modification of the properties of the coatings was achieved by subjecting the samples to heat treatment for two hours at the temperature of: 100, 200, 300 and 400 °C. It has been shown that the temperature of annealing, surface morphology, chemistry of oxidized surface, corrosion resistance and microhardness created quite a complex network of relationships.

- As a result of heat treatment, coatings were gradually covered by a layer of oxidized nickel species. XPS analyses showed that NiOOH and $Ni(OH)_2$ dominated among them. However, with the increase in annealing temperature, the share of these compounds began to decline in the face of the increasing share of NiO. This phenomenon intensified after exceeding the annealing temperature of 200 °C, when metallic nickel was evidently oxidizing to NiO while the content of compounds containing –OH groups did not decrease.

- After annealing of the coatings at 300 °C and 400 °C the previously observed spheroidal morphology of nickel disappeared, although single nano-sized plates embedded in a clearly granular layer were visible. It is very likely that after annealing at 300 °C and 400 °C, the surface layer was of considerable thickness and was mainly composed of NiO, with a very low $Ni(OH)_2$ content. On this basis, it can be assumed that the change of the main component of the oxidized nickel layer influenced its morphology and barrier properties. This, in turn, clearly translated into a deterioration of the corrosion resistance of Ni coatings annealed at 300 °C, and especially at 400 °C, during exposure in NaCl solution. For the coating annealed at 200 °C polarization resistance reached a value of ca. 2 $M\Omega$ $cm^2$, while at 400 °C it was only about 100 $k\Omega$ $cm^2$. The resistances determined from EIS measurements, 1.7 $M\Omega$ $cm^2$ and 18 $k\Omega$ $cm^2$, respectively, related to the oxidized surface layer also seemed to confirm this trend.

- The temperature of heat treatment of nickel coatings electrodeposited in DES bath largely influenced their microhardness. It has been shown that the observed behavior cannot be associated neither with changes in the surface composition nor the surface topography. Structural studies using the XRD technique showed that as the temperature of the heat treatment increased from 100 °C to 400 °C, the mean crystallite size increased from 13 to 35 nm (at 10 nm for the unannealed coating). Microhardness also changes—the maximum value of which was measured for the annealed coating at 200 °C.

In conclusion, the heat treatment of nickel coatings obtained from the eutectic mixture influenced both their microstructure as well as the mechanical and corrosive properties. The most promising results were obtained for the coating annealed at 200 °C, which manifested the highest microhardness and corrosion resistance.

**Supplementary Materials:** The following are available online at https://www.mdpi.com/article/10.3390/coatings11111347/s1, Figure S1: Nyquist plot of impedance spectra recorded after 168 h exposure in 0.05 mol $dm^{-3}$ NaCl for the as-deposited and heat treated nickel coatings for 2 h at: 100; 200; 300 and 400 °C.

**Author Contributions:** J.W.: Conceptualization, Methodology, Investigation, Project administration, Supervision, Resources, Funding acquisition, Writing—original draft. A.N.: Investigation, Writing—original draft. W.T.: Investigation, Methodology, Resources, Writing—original draft. K.W.: Investigation, Writing—original draft. K.P.: Investigation. B.S.: Resources, Writing—review and editing. All authors have read and agreed to the published version of the manuscript.

**Funding:** This research was funded by Polish Ministry of Science and Higher Education for the Department of Advanced Material Technologies (K26W03D05) at Wrocław University of Science and Technology in 2021 year, grant number 8211104160.

**Institutional Review Board Statement:** Not applicable.

**Informed Consent Statement:** Not applicable.

**Data Availability Statement:** The raw data required to reproduce these findings are available to anyone after sending an inquiry to juliusz.winiarski@pwr.edu.pl. The processed data required to reproduce these findings are available to anyone after sending an inquiry to juliusz.winiarski@pwr.edu.pl.

**Conflicts of Interest:** The authors declare no conflict of interest.

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
