# Peer review of "Effect of Annealing on Surface Morphology and Structure of Nickel Coatings Deposited from Deep Eutectic Solvents"

_coatings, doi:10.3390/coatings11111347_

Round 1

Reviewer 1 Report

The article entitled “Effect of annealing on surface morphology and structure of nickel coatings deposited from deep eutectic solvents” has examined the effect of heat treatment on characteristics and corrosion behavior of Ni coatings electrodeposited from DES baths. The article needs major revisions before it can be published. My opinions and questions are as follows:

  • Highlights?
  • Line 68 should be rewritten.
  • How have the authors selected the electrodeposition current density and deposition time? How much was the current efficiency?
  • How were the coatings with a thickness of 20µm produced for hardness measurements? May the variation of deposition conditions affect the hardness of these samples compared with the original ones? Why were not the thick coatings used for heat treatment and characterization too?
  • Lines 172 to 181 should be moved to the experimental section.
  • After heat treatment for 2h, the thickness of the formed oxide layer is not enough to be detected by XRD?
  • In lines 208 to 218 several subjects should be considered and corrected: 1- Your coatings are not microcrystalline. All of the coatings are nanocrystalline. Therefore, the relevant sentence is not related to your results. 2- All the samples were heat-treated for a constant duration. Thus, time cannot be an effective parameter in your results. I think the main and correct discussion has been done in the next paragraph (lines 222 to 237) and these sentences should be omitted.
  • Lines 218 to 221 should be omitted.
  • Could the authors see the morphological variations after the heat treatment in lower magnification micrographs? Adding lower magnification micrographs can be helpful.
  • Line 282: “De Los Santos Valladares et al. [3] presented a possible mechanism for the formation of a nickel oxide layer on a nickel coating. They indicated the diffusion of oxygen and nickel in the nickel oxide layer as a possible cause of voids formed in the layer.” Nickel oxide is a p-type oxide with cationic vacancies. The growth of this oxide layer may cause the gathering of the vacancies at the metal/oxide interface to form voids. According to the SEM micrographs, the voids are present throughout the cross-section of the heat-treated sample. Therefore, I think the proposed reason for the formation of voids is not a correct one and should be omitted.
  • The authors should add an SEM micrograph from the cross-section of the heat-treated samples that shows the formed oxide layer.
  • What is the reason for increasing Rp of 100°C heat-treated sample with time?
  • Can the authors explain what the positive Ecorr means?]
  • Nyquist plots should be added.
  • Add the fit results as continuous lines to bode plots. What is the problem of using CPE instead of pure capacitor in model III? CPE can be similar to a pure capacitor when its parameters change. CPE may result in a better fit for non-uniform surfaces.
  • How do the authors explain two times constant in the equivalent circuit of the as-deposited sample?
  • Why does the 200°C heat-treated sample have the highest Cdl? Can it be due to the large surface area in contact with corrosive media in this sample?
  • Conclusions should be summarized.

Author Response

The authors thank the Reviewers for valuable comments on the submitted manuscript. We made the necessary corrections and unify the description of the samples in all figures and tables. We hope the corrections will improve the article.

  1. Highlights?

Although highlights are not mandatory in the submission system, we have prepared them. The article is best described by the following highlights:

  • Structure and surface of Ni coatings modified through post-deposition heat treatment.
  • Domination of NiO over NiOOH and Ni(OH)2 with increasing annealing temperature.
  • Heat treatment from 100 to 400 °C increased mean crystallite size from 13 to 35 nm.
  • Maximum microhardness and corrosion resistance was reached after annealing at 200 °C.
  1. Line 68 should be rewritten.

The line indicated by the Reviewer was corrected to: ”Although the heat treatment influences coatings microhardness, the addition of an alloying component, e.g. tungsten, to the coating has an even greater impact on the above-mentioned property.

  1. How have the authors selected the electrodeposition current density and deposition time? How much was the current efficiency?

Electrodeposition parameters have beene selected basing on our exploratory research and previous publication [Winiarski, J.; Niciejewska, A.; Ryl, J.; Darowicki, K.; Baśladyńska, S.; Winiarska, K.; Szczygieł, B. Ni/cerium Molybdenum Oxide Hydrate Microflakes Composite Coatings Electrodeposited From Choline Chloride : Ethylene Glycol Deep Eutectic Solvent, Materials 2020, 13, 924. https://doi.org/10.3390/ma13040924.].

The current efficiency, calculated on the basis of charge passed, mass gain and coatings cross-section measurements, was estimated at ca. 65-75%. Missing information on deposition parameters and current efficiency of the process were supplemented in the text.

  1. How were the coatings with a thickness of 20µm produced for hardness measurements? May the variation of deposition conditions affect the hardness of these samples compared with the original ones? Why were not the thick coatings used for heat treatment and characterization too?

Coatings for microhardness measurements were produced at the same parameters, but the time was proportionally extended in order to obtain deposits suitable for this type of measurement. Of course, we do not exclude the fact that the microhardness measured on the cross-section of the coating may change depending on the distance of the indentation point from the coating / base material interface. To minimize the impact of this effect, indentation was always made in the middle of the coating thickness. Nickel coatings with a thickness of 7-10 μ are most often used, hence the choice of thickness for the remaining tests.

  1. Lines 172 to 181 should be moved to the experimental section.

The indicated paragraph, which described how the crystallite sizes were calculated, has been moved to the Experimental section (2.2 Research techniques).

  1. After heat treatment for 2h, the thickness of the formed oxide layer is not enough to be detected by XRD?

Indeed. The XRD technique is not predisposed to determining the structure of thin oxide layers. That’s why we used the XPS technique. An alternative would be to perform additional XRD analyzes in the grazing incidence mode, which we will certainly remember in the future publications.

  1. In lines 208 to 218 several subjects should be considered and corrected: 1- Your coatings are not microcrystalline. All of the coatings are nanocrystalline. Therefore, the relevant sentence is not related to your results. 2- All the samples were heat-treated for a constant duration. Thus, time cannot be an effective parameter in your results. I think the main and correct discussion has been done in the next paragraph (lines 222 to 237) and these sentences should be omitted.

The inaccuracies raised by Reviewer have been corrected in the revised text: 1. “Increase in the size of the crystallites is often associated with an increase in microhardness.” instead „The transition from nanocrystalline to microcrystalline structure is often associated with an increase in microhardness.” Indeed, looking through the size of the crystallites (<100nm), all the coatings were nanocrystalline. However, in the literature one can also find another distinction between micro- and nanocrystalline materials resulting from a sudden / abrupt change in material properties. We found this change of properties (microhardness, surface chemistry and corrosion resistance) and hence the use of the term “microcrystalline”. However, wanting to refer only to the size in the revised text we used “nanocrystalline” for all produced coatings. 2According to the Reviewer's remark, time was not a variable in the annealing process, so this fragment was removed from the text.

  1. Lines 218 to 221 should be omitted.

This fragment has been removed from the manuscript.

  1. Could the authors see the morphological variations after the heat treatment in lower magnification micrographs? Adding lower magnification micrographs can be helpful.

New drawings (Figs 3a-3e) were made. They contain lower magnification photos (about 25,000x) as background with inserts showing the surface features at high magnification (about 100,000x).

  1. Line 282: “De Los Santos Valladares et al. [3] presented a possible mechanism for the formation of a nickel oxide layer on a nickel coating. They indicated the diffusion of oxygen and nickel in the nickel oxide layer as a possible cause of voids formed in the layer.” Nickel oxide is a p-type oxide with cationic vacancies. The growth of this oxide layer may cause the gathering of the vacancies at the metal/oxide interface to form voids. According to the SEM micrographs, the voids are present throughout the cross-section of the heat-treated sample. Therefore, I think the proposed reason for the formation of voids is not a correct one and should be omitted.

The authors agree with the Reviewer that the cited fragment does not explain the formation of empty spaces in the microstructure of the nickel coating, but the preservation of the nickel oxide layers during heat treatment. Consequently, an inadequate text fragment has been corrected.

  1. The authors should add an SEM micrograph from the cross-section of the heat-treated samples that shows the formed oxide layer.

We tried to image one of these layers using HR-SEM, but even after cross-sectioning with the FIB technique, the thickness of this layer was very small and it did not allow obtaining sharp images that would add any value to this article. Therefore we have decide to investigate the morphology and chemistry of surface layers only by XPS technique.

  1. What is the reason for increasing Rp of 100°C heat-treated sample with time?

This is a very good question. Unfortunately, the answer to it is not easy. There may be several reasons, like: oxidized layer composition (higher NiOOH / NiO ratio than for the other layers), different morphology / porosity and so on. We have only managed to point out some possible causes in this article.

  1. Can the authors explain what the positive Ecorr means?

We used, for example, the phrase: “coatings after heat treatment were characterized by more positive corrosion potentials”. “More positive” in this context means: shifted towards positive values of Ecorr. This is a popular phrase that is often used in the literature. However, for clarity, this sentence has been changed and now it sounds: ”It was noticed that the corrosion potentials of the coatings after heat treatment were higher (by more than 300 mV) than Ecorr of the as-deposited Ni coating.”

  1. Nyquist plots should be added.

Nyquist and Bode representations of the impedance spectra are complementary. We chose the Bode diagram because it gave much more information to the article for these specific results. First of all, it shows both time constants and their shift as a function of heat-treatment temperature, which was completely absent on the Nyquist plot. Secondly, we met with critical comments from reviewers many times, so as not to duplicate the results and present the impedance spectra only in the one selected notation.

  1. Add the fit results as continuous lines to bode plots. What is the problem of using CPE instead of pure capacitor in model III? CPE can be similar to a pure capacitor when its parameters change. CPE may result in a better fit for non-uniform surfaces.

Fitting lines have been added to Figs 9a and 9b. Regarding the use of CPE – we agree that the use of CPE usually gives a better fit as it takes into account the imperfections (non-uniform distribution of time constants, uneven distribution of reaction kinetics, etc.) of the corroding surface. When choosing models and circuit elements, we were also guided by another rule, which said that the simpler the elements we use, the easier it will be to explain their physical meaning. This was also the case with the use of capacitor instead the CPE element in model III. We treated Model II (with CPEdl) only as a starting point. Since it turned out that the alpha exponent was close to one, we allowed ourselves to use the pure capacitor element here. Thanks to this, we got straight true capacitance without the need to calculate Cdl, and without noticeable deterioration of the fitting. Unfortunately, the surface of the sample annealed at 400 °C was already so heterogeneous that the further use of pure capacitor was impossible. Therefore, Model II with CPE was used for this coating, and the Cdl value was calculated from Eq. 4.

  1. How do the authors explain two times constant in the equivalent circuit of the as-deposited sample?

The surface of the as-deposited nickel coating was also covered with an "oxide" layer, but much less thick. Therefore, also here, in the EIS research, the existence of a certain surface layer was taken into account, the electric capacity and electrolytic resistance of which in the pores give a frequency response in the form of a separate time constant. This is one of the approaches commonly used in the literature.

  1. Why does the 200°C heat-treated sample have the highest Cdl? Can it be due to the large surface area in contact with corrosive media in this sample?

If Cdl were related to the porosity of the oxide layer, we could connected this high value to thicker double layer or to the existence of larger number of pores connected in parallel that resulted in higher electrical capacitance. 

  1. Conclusions should be summarized.

This section have been corrected and a sentence summarizing the conclusions presented in points has been added: “In conclusion, the heat treatment of nickel coatings obtained from the eutectic mixture influences both their microstructure as well as the mechanical and corrosive properties. The most promising results were obtained for the coating annealed at 200 °C, which manifested the highest microhardness and corrosion resistance.

Reviewer 2 Report

The article Effect of annealing on surface morphology and structure of nickel coatings deposited from deep eutectic solvents describes nickel coatings deposited on a copper base material from DES and heat treatment at temperatures from 100 to 400 °C.

The abstract is prepared concisely and presents the most critical aspects of the research conducted.

The Introduction section clearly presents the theoretical background of the research undertaken. The purpose of the research has been well motivated.

The materials and methods section explicitly describes used research methods. However, after reviewing the entire article, I think it needs a slight reorganization. First, I would move the description of the XPS method before the XRD method. Knowing the chemical composition makes it easier to analyze, and I would do the same in the Results section. Results and discussion lines 172-185 In the lines listed, there is a description of the Scherrer method used to calculate the size of the crystallites. It seems to me that it is reasonable to place this description in the methodology section since it is directly related to it.  This also applies to the description of the corrosion testing methodology from line 432.

Perhaps it would be helpful to include a table explaining the sample descriptions in, for example, fig 1. In this figure, enter the markings used further in the text.

For future reference, in further research, I would suggest using Grazing Incidence X-ray Diffraction (GIXRD), which works well for examining layers.

The results and discussion section is clearly written and gives relevant results. I would consider moving the XPS results ahead of the XRD results. This seems reasonable to me because it clearly shows learning the chemical and then phases composition of the resulting material. The authors included elements of discussion and literature references.

Table 1. I understand that the authors want to show the correlation of crystallite size with mechanical properties, but it would be clearer to separate these values somehow.

The research is well designed and presented. The manuscript as a whole needs minor revisions, but these are mainly organizational issues.

Congratulations on an interesting article.

Author Response

The authors thank the Reviewers for valuable comments on the submitted manuscript. We made the necessary corrections and unify the description of the samples in all figures and tables. We hope the corrections will improve the article.

Responses to the reviews

  1. The materials and methods section explicitly describes used research methods. However, after reviewing the entire article, I think it needs a slight reorganization. First, I would move the description of the XPS method before the XRD method. Knowing the chemical composition makes it easier to analyze, and I would do the same in the Results section. Results and discussion lines 172-185 In the lines listed, there is a description of the Scherrer method used to calculate the size of the crystallites. It seems to me that it is reasonable to place this description in the methodology section since it is directly related to it.  This also applies to the description of the corrosion testing methodology from line 432.

The method of presenting the research results proposed by the authors is not accidental. At the beginning, the volumetric properties of the obtained coatings are characterized, i.e. the phase composition, the size of the crystallites and the related microstructure and microhardness. In the next stage, the results of XPS are discussed, concerning the thin layer on the surface of the coatings, which also affects the corrosion properties of the described layers.

The lines 172-185, which described how the crystallite sizes were calculated, has been moved to the Experimental section (2.2 Research techniques).

  1. Perhaps it would be helpful to include a table explaining the sample descriptions in, for example, fig 1. In this figure, enter the markings used further in the text.

The description of the samples has been corrected and unify in the tables and figures of the manuscript.

  1. For future reference, in further research, I would suggest using Grazing Incidence X-ray Diffraction (GIXRD), which works well for examining layers.

We agree with the Reviewer that grazing incidence mode would be good for studying the structure of oxidized nickel layers and would undoubtedly be a valuable supplement to the XPS technique. We know the proposed method and we have already used it in our research [B. Szczygieł, A. Laszczyńska, W. Tylus, Influence of molybdenum on properties of Zn–Ni and Zn–Co alloy coatings, Surface and Coatings Technology, Volume 204, Issues 9–10, 2010, Pages 1438-1444, https://doi.org/10.1016/j.surfcoat.2009.09.042.]. We will definitely try to use it again in the next research depending on its availability.

  1. The results and discussion section is clearly written and gives relevant results. I would consider moving the XPS results ahead of the XRD results. This seems reasonable to me because it clearly shows learning the chemical and then phases composition of the resulting material. The authors included elements of discussion and literature references.

As mentioned, the order of the article was organized from the description of the volumetric properties to the factors resulting from changes in the surface of the layer.

5.Table 1. I understand that the authors want to show the correlation of crystallite size with mechanical properties, but it would be clearer to separate these values somehow.

Unfortunately, creating a separate table for only one parameter could be considered an unnecessary expansion of the volume of the article.

Reviewer 3 Report

Manuscript devoted to investigation of effect of annealing on surface morphology and structure of nickel coatings deposited from deep eutectic solvents. Its interesting and actual research. But there are some comments on the paper.

In the Introduction it should be noted that nickel can be used as a matrix of composite electrochemical coatings and refer to the following publications:

  1. Gobinda Gyawali, Bhupendra Joshi, Khagendra Tripathi, Soo Wohn Lee. Effect of ultrasonic nanocrystal surface modification on properties of electrodeposited Ni and Ni-SiC composite coatings // Journal of Materials Engeneerig and Performance. 2017. V. 26. P. 4462 – 4469.

2. Lanzutti A., Lekka M., de Leitenburg C., Fedrizzi L. Effect of pulse current on wear behaviour of Ni matrix micro- and nano-SiC composite coatings at room and elevated temperature. Tribology International. 2019. V. 132. P. 50 – 61.

3. Tseluikin V.N., Vasilenko E.A. Electrodeposition and properties of composite coatings based on nickel // Russian Journal of Applied Chemistry. 2011. V. 84. № 11. P. 2005 – 2007.

Author Response

Please see the Word.

Reviewer 4 Report

The influence of different heat treatments on the surface of Ni coatings were investigated by different analysis techniques. There are still minor things need to think about it or to be changed.

Keywords (Line 26-27)

Suggested to give as follows:-

Nickel coatings, heat treatment/annealing temperature, microhardness, porosities, corrosion resistance, X-ray analysis or electron microscopy.

  1. Introduction (Line 29-88)

  It looks fine

2. Materials and methods

2.1 materials... (Line 89-110)

At line 110, Why did samples not quench in water to freeze the annealed microstructure? could the authors give cooling rate?.

2.2 research .. (line 117-163)

At line 117, could the authors give the penetration depth of the investigated coatings surface if AV 2KV was used?

At lines 127-129, were the samples etched before the microhardness values were measured? Are there any observed particles at the investigated area?

3. Results and discussion (line 164-497)

Between line 175 and line 181, no standard deviation (SD) for data analysis was given. In addition, SD should also given for crystallite size (D) in Table 1.

Below Table 1, () should give instead of *.

At line 198, ..was only slightly higher than ..

It could specify here percentage, like A-system was % higher than system B.

At line 204, ..to a value lower ...

Se above comment, A-system was % lower than B-system

At lines 222-237, Could the authors use EBSD to find out the cause.

On Figure 2, could the authors estimate the size of voids and provide it on it.?

At line 247, could the authors show particles (nanoplates..) on the image in Figure 3a-3c).

Line 271-291, can fine agglomerates of crystallites/voids be shown on the figures.

Could the authors specify how large the size of voids and describe the shape, depth and so on.

could the authors take EBSD maps for different annealing temperatures to see the strain/stress field in all structures, in particular at 200 °C, 300 °C, and 400 °C?

Figure 5 between line 310 and line 315, There is no information on the y-axis.

The same thing on  Figure 6, please se above comment. What did mean with the peaks.

Line 353-401:- Could chloride (Cl)  in pores on the surface facilitates the initiation of corrosion.

How look pores (form), stress field round them, where are the pores, if they are mainly in the grain boundaries, those should be weakened , and leading to decline the corrosion resistance. If the material have a lot of pores on the surface, the passive film will develop inhomogenously.

Could the author give some citations that the quality of the passive film deteriorates at a more uneven substrate surface, probably shallow pits with rounded edges give less effect.

From line 403 to line 495,  the EIS can only show the passivity of the whole measured surface by itself, without compare with a perfect surface, how could you directly to use the EIS data?.

Author Response

1、Introduction (Line 29-88). It looks fine.

Thank you.

2、Materials and methods. 2.1 materials... (Line 89-110). At line 110, Why did samples not quench in water to freeze the annealed microstructure? could the authors give cooling rate?

The samples were not frozen. The standard method of free cooling of the samples in the furnace was used, as is the case with most annealing processes of Ni or Cr coatings.

3、2.2 research .. (line 117-163). At line 117, could the authors give the penetration depth of the investigated coatings surface if AV 2KV was used?

It is difficult for us to estimate the penetration depth of the material during the imaging. The accelerating voltage applied was small, so the sampling depth was also very small. We wanted to visualize the surface morphology as faithfully as possible, and as it is well known, this is achievable at low accelerating voltages.

4、At lines 127-129, were the samples etched before the microhardness values were measured? Are there any observed particles at the investigated area?

The samples were not etched. No particles were observed in the study area.

5、3. Results and discussion (line 164-497). Between line 175 and line 181, no standard deviation (SD) for data analysis was given. In addition, SD should also given for crystallite size (D) in Table 1.

Due to the intensity and symmetry, the Ni (111) peak was selected for the crystallite size calculations. The diffraction pattern (Figure 1) shows that the peaks above 70° are not symmetrical and could not be taken into account. Usually, peaks from the range of 2θ: 30° to 50° are used for the calculation of crystallite size. For high angles, the instrumental factors and lattice stresses overlap the peak width. It should also be highlighted that the tested Ni coatings are the polycrystalline material in which the actual crystals can be composed of any number of domains that are involved in the X-ray scattering phenomenon. Often, well-defined shape crystallites may not be homogeneous with respect to their size. So the obtained value expresses the average value of the crystallites. Hence the expression “average” in the text, but it is not an arithmetic “average” resulting from averaging the results, namely the size of the crystallites calculated for peaks at different 2θ. To avoid confusion, the expression “average” has been removed from the main text.

6、Below Table 1, () should give instead of *.

Thank you for your attention. Corrected.

7、At line 198, ..was only slightly higher than ..It could specify here percentage, like A-system was % higher than system B.

Indeed, the sentence we use may not be entirely precise. The sentence: “Initially, for the as-deposited Ni coating, the indentation microhardness amounted to 589 Vickers (Table 1) which was only slightly higher than those (425-500 HV) meas-ured by Danilov et al. [17] for similar Ni coatings deposited in DES and these measured by Abraham et al. [18] for coatings from conventional Watt’s plating baths (517-560 HV).” has been changed to: “Initially, for the as-deposited Ni coating, the indentation microhardness amounted to 589 Vickers (Table 1) which was 18-38% higher than those measured by Danilov et al. [17] for similar Ni coatings deposited in DES (425-500 HV) and these measured by Abraham et al. [18] for coatings from conventional Watt’s plating baths (517-560 HV).”

8、At line 204, ..to a value lower ... Se above comment, A-system was % lower than B-system

Indeed, the sentence we use may not be entirely precise. The sentence :”Further increase in the heating temperature up to 400 °C was not favorable, because a decrease in microhardness to a value lower (475 Vickers) than that calculated for as-deposited Ni (589 Vickers) was observed (Table 1).” has been changed to: “Further increase in the heating temperature up to 400 °C was not favorable, because a decrease in microhardness to 475 Vickers was observed (Table 1).”

9、At lines 222-237, Could the authors use EBSD to find out the cause.

Indeed, we could use the EBSD method. The suggestion is very accurate and we will certainly use this method in the next research.

10、On Figure 2, could the authors estimate the size of voids and provide it on it.?

Determining such a relationship and finding a correlation between mean crystallite size, microhardness and temperature would be very valuable, but would require a separate and quite extensive experiment for the entire series of samples. We are considering performing such tests in the near future.

11、At line 247, could the authors show particles (nanoplates..) on the image in Figure 3a-3c).

In our opinion, these nanoplates are very well visible in the pictures and we would like to avoid adding more graphics in the pictures.

12、Line 271-291, can fine agglomerates of crystallites/voids be shown on the figures.

In Figs 4b and 4d, the markers for exemplary voids and agglomerates of crystallites have been added.

13、Could the authors specify how large the size of voids and describe the shape, depth and so on.

The sentence: “The size of the voids ranged from single nanometers to 20 nm for the as-deposited Ni coating, while the coating annealed at 300 °C was characterized by voids even up to 80 nm in diameter.” has been added to the text.

14、could the authors take EBSD maps for different annealing temperatures to see the strain/stress field in all structures, in particular at 200 °C, 300 °C, and 400 °C?

Unfortunately, this would require additional and costly research, including ion beam etching of the coatings. We will certainly consider this approach in future studies.

15、Figure 5 between line 310 and line 315, There is no information on the y-axis.

When presenting the XPS spectra in this way, the label on the Y axis is usually not given, and the possible specification of arbitrary units (a.u.) will not contribute anything to the interpretation when we compare the shape of the spectrum with each other.

16、The same thing on Figure 6, please se above comment. What did mean with the peaks.

When presenting the XPS spectra in this way, the label on the Y axis is usually not given, and the possible specification of arbitrary units (a.u.) will not contribute anything to the interpretation when we compare the shape of the spectrum with each other. Regarding the peaks – due to a very complicated character of the presented Ni 2p spectra and a large number of spectral components (and the degree of complication of individual spectral components – Ni, NiO, NiOOH, Ni(OH)2 – their spectra are not a single peaks, but often dublets with satellite peaks), it is difficult to add  peak descriptions in the drawings. The entire complicated deconvolution procedure was taken from the works of Biesinger et al. [M.C. Biesinger, B.P. Payne, L.W.M. Lau, A. Gerson, R.St.C. Smart, X-ray photoelectron spectroscopic chemical state quantification of mixed nickel metal, oxide and hydroxide systems, Surf. Interface Anal. 41 (2009) 324–332., A.P. Grosvenor, M.C. Biesinger, R.St.C. Smart, N.S. McIntrye, New interpretations of XPS spectra of nickel metal and oxides, Surf. Sci. 600 (2006) 1771–1779., M.C Biesinger, B.P. Payne, A.P. Grosvenor, L.W.M. Lau, A.R. Gerson, R.St.C. Smart, Resolving surface chemical states in XPS analysis of first row transition metals, oxides and hydroxides: Cr, Mn, Fe, Co and Ni, Appl. Surf. Sci. 257 (2011) 2717–2730., M.C Biesinger, L.W.M. Lau, A.R. Gerson, R.St.C. Smart, The role of the Auger parameter in XPS studies of nickel metal, halides and oxides, Phys. Chem. Chem. Phys. 14 (2012) 2434–2442.], then adapted to the spectra and repeatedly used in our previous research on Ni and Ni alloy coatings [A. Laszczyńska, J. Winiarski, B. Szczygieł, I. Szczygieł, Electrodeposition and characterization of Ni–Mo–ZrO2 composite coatings, Applied Surface Science, Volume 369, 2016, Pages 224-231., A. Laszczyńska, W. Tylus, J. Winiarski, I. Szczygieł, Evolution of corrosion resistance and passive film properties of Ni-Mo alloy coatings during exposure to 0.5M NaCl solution, Surface and Coatings Technology, Volume 317, 2017, Pages 26-37., A. Laszczyńska, W. Tylus, B. Szczygieł, I. Szczygieł, Influence of post−deposition heat treatment on the properties of electrodeposited Ni−Mo alloy coatings, Applied Surface Science, Volume 462, 2018, Pages 432-443., A. Laszczyńska, W. Tylus, I. Szczygieł, Electrocatalytic properties for the hydrogen evolution of the electrodeposited Ni–Mo/WC composites, International Journal of Hydrogen Energy, Volume 46, Issue 44, 2021., A. Laszczyńska, I. Szczygieł, Electrocatalytic activity for the hydrogen evolution of the electrodeposited Co–Ni–Mo, Co–Ni and Co–Mo alloy coatings, International Journal of Hydrogen Energy, Volume 45, Issue 1, 2020, Pages 508-520., Juliusz Winiarski, Beata Cieślikowska, Włodzimierz Tylus, Piotr Kunicki, Bogdan Szczygieł, Corrosion of nanocrystalline nickel coatings electrodeposited from choline chloride:ethylene glycol deep eutectic solvent exposed in 0.05 M NaCl solution, Applied Surface Science, Volume 470, 2019, Pages 331-339.].

17、Line 353-401:- Could chloride (Cl) in pores on the surface facilitates the initiation of corrosion.

Chloride ions usually initiate local corrosion of passivating metals.

18、How look pores (form), stress field round them, where are the pores, if they are mainly in the grain boundaries, those should be weakened , and leading to decline the corrosion resistance. If the material have a lot of pores on the surface, the passive film will develop inhomogenously.

We are unable to define it. However, we agree with the reviewer that a large number of pores will certainly cause heterogeneity in the created passive film.

19、Could the author give some citations that the quality of the passive film deteriorates at a more uneven substrate surface, probably shallow pits with rounded edges give less effect.

The shape of the pores, their diameter, depth and other geometrical parameters without no doubt influence the electrolytic resistance of corrosive solution  passing them, and thus the electrical capacitances connected with this layer and the double layer capacitance that forms at the bottom of the pores at metal/electrolyte interface. 

20、From line 403 to line 495, the EIS can only show the passivity of the whole measured surface by itself, without compare with a perfect surface, how could you directly to use the EIS data?

In our measurements the EIS technique is undoubtedly a global technique that brings the information of the sum of all processes taking place on the surface of corroding metal being in contact with NaCl electrolytic solution. Fortunately, thanks to the use of electric equivalent circuits and some physical models we are able to split the resistances and capacitances to these related to individual electrochemical and physical processes. In any case, at least we can draw some conclusions about the variation of surface state being in contact with the corrosive electrolyte.

Round 2

Reviewer 1 Report

The authors have modified the manuscript according to the comments. There are only three points that have not been well considered. The manuscript can be accepted after minor modifications.

  • Can the authors explain what the positive Ecorr means? I did not mean more positive values. My question is what is the meaning of, for example, +150 mV vs. Ag/AgCl in Fig. 8. Can the authors explain the corrosion reactions?
  • Nyquist plots should be added. It is easier to compare the Rp values in the Nyquist plots.
  • Why does the 200°C heat-treated sample have the highest Cdl? Can it be due to the large surface area in contact with corrosive media in this sample? Answer: “If Cdl were related to the porosity of the oxide layer, we could connected this high value to thicker double layer or to the existence of larger number of pores connected in parallel that resulted in higher electrical capacitance.” Question: What can cause a thicker double layer in this sample? Does a thicker double layer decrease the capacitance or increase it?
